# Cardiac Imaging in Anderson-Fabry Disease: Past, Present and Future

**DOI:** 10.3390/jcm10091994

**Published:** 2021-05-06

**Authors:** Roberta Esposito, Ciro Santoro, Giulia Elena Mandoli, Vittoria Cuomo, Regina Sorrentino, Lucia La Mura, Maria Concetta Pastore, Francesco Bandera, Flavio D’Ascenzi, Alessandro Malagoli, Giovanni Benfari, Antonello D’Andrea, Matteo Cameli

**Affiliations:** 1Department of Clinical Medicine and Surgery, Federico II University Hospital, 80122 Naples, Italy; vittycuomo@yahoo.it; 2Mediterranea Cardiocentro, 80122 Naples, Italy; 3Department of Advanced Biomedical Sciences, Federico II University Hospital, 80131 Naples, Italy; ciro.santoro@unina.it (C.S.); rejinasorrentino@gmail.com (R.S.); lucia.lamura@hotmail.it (L.L.M.); 4Division of Cardiology, Department of Medical Biotechnologies, University of Siena, 53100 Siena, Italy; giulia_elena@hotmail.it (G.E.M.); pastore2411@gmail.com (M.C.P.); flavio.dascenzi@unisi.it (F.D.); matteo.cameli@yahoo.com (M.C.); 5Heart Failure Unit, Department of Cardiology, IRCCS Policlinico San Donato Milanese, University of Milan, 20122 Milan, Italy; francescobandera@gmail.com; 6Nephro-Cardiovascular Department Division of Cardiology, Baggiovara Hospital, University of Modena and Reggio Emilia, 41121 Modena, Italy; ale.malagoli@gmail.com; 7Division of Cardiology, Department of Medicine, University of Verona, 37132 Verona, Italy; giovanni.benfari@gmail.com; 8Division of Cardiology, Umberto I Hospital Nocera Inferiore (Salerno), Luigi Vanvitelli University, 81100 Caserta, Italy; antonellodandrea@libero.it

**Keywords:** Anderson-Fabry disease, echocardiography, cardiac involvement, cardiac imaging, multimodality imaging

## Abstract

Anderson-Fabrydisease is an X-linked lysosomal storage disorder caused by a deficiency in the lysosomal enzyme α-galactosidase A. This results in pathological accumulation of glycosphingolipids in several tissues and multi-organ progressive dysfunction. The typical clinical phenotype of Anderson-Fabry cardiomyopathy is progressive hypertrophic cardiomyopathy associated with rhythm and conduction disturbances. Cardiac imaging plays a key role in the evaluation and management of Anderson-Fabry disease patients. The present review highlights the value and perspectives of standard and advanced cardiovascular imaging in Anderson-Fabry disease.

## 1. Introduction

Anderson-Fabry disease (AFD) is classified as an X-linked storage disorder caused by the abnormal activity of a lysosomal enzyme called α-galactosidase A. AFD results in a pathological accumulation of glycosphingolipids in several tissues that generates different disease phenotypes according to the extent and severity of the involved organ [1,2,3] Renal failure, cardiomyopathy, as well as peripheral and central nervous system involvement are the leading causes of morbidity in these patients [1]. Cardiomyopathy is the leading cause of death in AFD, accounting for 38% of all-cause mortality [4].

The prevalence of AFD varies depending on the screening method used. Neonatal screening programs reported an unexpectedly high incidence ranging from 1:1250 to 1:7800 [5,6]. Because of its X-linked recessive inheritance, female carriers can exhibit mild-to-severe symptoms due to variable expression according to random X inactivation (“Lyonization”) of the affected gene in embryogenesis. The typical clinical phenotype of AFD cardiomyopathy is progressive hypertrophic cardiomyopathy associated with heart failure, rhythm and conduction disturbances [7,8,9,10]. Glycosphingolipids accumulation in the myocyte macroscopically expresses itself as left ventricular hypertrophy (LVH) in early phases and myocardial fibrosis replacement in the late disease stages [9,10,11]. AFD cardiomyopathy can mimic clinical and structural features of hypertrophic cardiomyopathy (HCM) [9,10,11,12,13]. Its prevalence among patients presenting with late-onset HCM is estimated to be 6% among men and 12% among women [14,15]. Early diagnosis of AFD cardiomyopathy is crucial because AFD is treatable with disease-specific treatments, such as enzyme replacement therapy (ERT) or chaperone therapy with Migalastat [16,17,18,19]. Timely ERT initiation is crucial, since in patients with advanced Fabry disease, it does not seem to prevent progression towards severe organ failure and death [20].

Therefore, prompt diagnosis and treatment of patients with Fabry disease seem warranted.

LVH should always be evaluated together with family history and extracardiac symptoms. LVH is a more common finding in AFD male patients than female (53% versus 33% in untreated patients) [10]. In males, LVH occurs at a younger age (42 vs. 50.1 years) than in females [21]. Careful medical history collection and thorough physical examination are mandatory in the setting of LVH. The presence of characteristic symptoms and signs, such as corneal opacity, angiokeratoma, hypohidrosis, albuminuria andacroparesthesia, in a patient with LVH should immediately raise the suspicion of AFD. However, in some rare cases, the heart may be the only organ involved making the diagnosis of AFD challenging even with sophisticated imaging techniques.

## 2. Standard and Advanced Echocardiography

Standard echocardiography (Table 1) is the first-line imaging modality for the identification of the typical features of AFD cardiomyopathy, such as unexplained left ventricular hypertrophy, preserved left ventricular ejection fraction (LVEF) until end-stage of the disease, progressive diastolic dysfunction and right ventricular (RV) free wall thickening [22,23]. However, these findings are not specific to AFD, as they may occur in other types of left ventricular hypertrophy.

### 2.1. Left Ventricular Morphology and Systolic Function

AFD cardiomyopathy is characterized by concentric remodeling usually progressing to concentric hypertrophy [24]. In AFD patients, LVH is most commonly homogeneously distributed, as opposed to asymmetric HCM (Figure 1A), and only rarely occurs as asymmetric septal hypertrophy or eccentric hypertrophy [24]. LV outflow tract obstruction has a very low incidence at rest but is reported in 43% of patients during effort [25]. LVEF is usually preserved or even supranormal in the early stages of cardiac involvement [26]. LVEF reduction, in the advanced phases of the disease, is correlated to a worse prognosis [7]. In some patients, the posterior and inferior LV wall may appear hypokinetic or akinetic as an expression of myocardial fibrosis.

In the past, the echocardiographic binary appearance of the LV endocardial border, namely the binary sign, was considered the pathognomonic sign to discriminate patients suffering from AFD from those with familial HCM [27]. This sign consists of a hyperechogenic endocardial surface and hypoechogenic subendocardial layer (Figure 1B). This echocardiographic finding reflects the endocardial and subendocardial glycosphingolipids compartmentalization. More recent studies have reduced the relevance of this sign because of its very low sensitivity and specificity [28,29]. Another characteristic feature of AFD cardiomyopathy is the presence of prominent papillary muscles (Figure 1C,D) [30]. However, this is not an early sign and, therefore, should not be used for screening purposes.

### 2.2. Left Ventricular Diastolic Function

LV diastolic dysfunction occurs early in the AFD disease stages (Figure 2) and represents the substrate for the onset of symptoms and the leading cause of heart failure in these patients [31]. However, LV diastolic dysfunction is not usually severe in AFD. In a report by Palecek et al., 44% of 81 echocardiographic examinations in 35 patients with Fabry disease had a normal LV filling pattern, while 63% showed impaired LV diastolic function. Of these, only 4% had a restrictive filling pattern, while 60% had a pseudo-normal filling pattern, and 36% impaired relaxation [32].

Diastolic dysfunction in AFD cardiomyopathy is related to progressive myocardial wall thickening. Its severity increases in the advanced stages of LVH [33,34,35]. However, compromised diastolic function, evaluated with tissue Doppler analysis, can anticipate LVH. Thus, the appearance of reduced TDI velocities might be the initial finding of cardiac involvement in AFD [35,36]. Furthermore, unlike LVEF, the severity of LV diastolic dysfunction is closely related to the NYHA class severity [37].

Diastolic function indices, such as mitral flow Doppler parameters and the Tei index, a marker for combined diastolic and systolic function, are often altered in AFD patients with LVH [38]. The Tei index does not provide any additional diagnostic contribution compared to TDI in the evaluation of ERT response [31]. In AFD patients with preserved LVEF, there is an association between diastolic dysfunction indices and the presence of late gadolinium enhancement (LGE) detected by cardiac magnetic resonance (CMR) imaging. In particular, a cut-off of 14.8 for the septal E/e’ ratio has shown to be the best predictor of the presence of LGE [39].

Diastolic dysfunction leads to the enlargement of the left atrium. In AFD, left atrial dilation occurs in the early stages of Fabry disease, even before the development of LVH [39,40,41]. Atrial damage may be the consequence of the accumulation of glycosphingolipids that has also been reported in atrial myocytes [42].

This suggests that the evaluation of atrial damage may be useful in the early diagnosis of AFD.

In most cases, atrial dilatation occurs as mild or moderate, becoming severe only in the presence of significant mitral valve disease, myocardial fibrosis and major LVH [43,44].

### 2.3. Right Ventricle

Right ventricular hypertrophy (RVH) (Figure 3) is frequent in AFD patients and correlates with LVH and disease severity in most cases. In contrast to LVH, which is more prevalent in men, RVH prevalence is similar in men and women and is estimated at ~40–70% of all patients with Fabry disease, and its prevalence increases with age [45,46]. RV systolic function is preserved in about half of the cases of AFD, even in the presence of RVH. Thus, the assessment of TAPSE, a parameter for global right ventricular function, is not useful for the evaluation of right ventricular involvement in AFD [46]. RV diastolic dysfunction is detectable in about half of the cases of AFD [45,47].

### 2.4. Aorta

In AFD, a remodeling process may affect the aorta. Barbey et al. [37] found aortic dilation at the sinus of Valsalva and ascending aorta in 32.7% of males and 5.6% of females and aneurysms in 9.6% of males and 1.9% of females among 106 patients with AFD, who underwent transthoracic echocardiography.

### 2.5. Valve Disease

The process of accumulation of glycosphingolipids may involve the valvular apparatus [48]. A thickening of the valve leaflets is present in about 25% of patients [49]. The most affected valves are the aortic and mitral valves [50]. Mitral valve insufficiency may also be induced by papillary muscle prominence. Valvular disease is usually mild, and severe valvular disease is only detectable when other contributing causes are present [44]. In a large series of AFD patients, valve disease was reported in 14.6% of patients (17% in males, 12% in females), but only 0.4% of cases were referred to surgical correction [25].

## 3. Speckle Tracking Echocardiography

Speckle tracking echocardiography (STE) (Table 2) has an incremental value in differentiating between primary and secondary LVH and in the differential diagnosis with storage diseases [51]. Moreover, STE enables early detection of intrinsic myocardial dysfunction before LVEF reduction [51]. Shanks et al. demonstrated that strain and strain rate (SR) analysis are useful in identifying AFD patients with reduced myocardial function, independently of LVH [52]. They found that longitudinal systolic strain and diastolic isovolumic SR were more accuratethan the other conventional echocardiographic measurements of myocardial contraction and relaxation. In AFD cardiomyopathy, STE-derived LV involvement appears to be complex and heterogeneous (Figure 4). Our group identified four distinct patterns of longitudinal strain (LS) impairment: the first with a normal or near-normal regional LS, the second with LS reduction in septal and anterior regions, the third with LS reduction in both septal/anterior and inferolateral regions and the fourth with LS reduction in the inferolateral region. However, all patients had a greater impairment of the basal segments compared to the apical segments of the left ventricle (Figure 4) [53]. The longitudinal strain bull’s eye pattern is markedly different between patients affected by HCM and AFD. In HCM patients, the involved region with reduced longitudinal strain is mainly located in the septum, while in late-stage AFD patients, it is located at the lateral and posterior walls [54]. Interestingly, the base-to-apex circumferential strain (CS) gradient is also different between patients affected by nonobstructive HCM and AFD. The preservation of base-to-apex circumferential strain gradient in non-obstructive HCM, but not in AFD, may help to differentiate these two cardiomyopathies from an echocardiography perspective [55]. In addition to the impairment of global LS and base-to-apex CS gradient, a reduced global CS can also differentiate AFD patients with LVH from those affected by HCM, who show an increased global CS [55].

The reduction in the inferolateral region strain has shown to be the most accurate parameter to identify the presence of LGE on CMR [43]. The possibility to indirectly predict LGE, i.e., myocardial fibrosis, using STE may be highly relevant in patients with contraindications to CMR, such as implanted devices, end-stage renal disease, considering that fibrosis has important prognostic implications in AFD.

STE advancements now make available the analysis of layer-specific myocardial deformation. In newly diagnosed, untreated AFD patients, our group demonstrated that layer-specific strain imaging highlights an impairment of LV longitudinal deformation, mainly involving subepicardial strain and causing an increase in LS myocardial gradient. These findings could be useful for identifying the mechanisms underlying early LV dysfunction in AFD patients [56].

Furthermore, there is evidence that STE-derived left atrial peak longitudinal deformation and RVLS are impaired in AFD patients compared to healthy controls, even when standard echocardiographic parameters are in the normal range [57,58].

Mechanical dispersion assessed with STE was observed by Cianciulli et al. in 76% of patients with AFD and LVH, while it was not evident when LVH was absent [59]. The use of mechanical dispersion in strain echocardiography is a recognized prognosticator of ventricular arrhythmias and HF [60,61], thus it may be beneficial to stratify the risk of disease progression and the need for cardiac resynchronization therapy in AFD.

## 4. Three-Dimensional Echocardiography

Three-dimensional echocardiography has some advantages over standard two-dimensional echocardiography. It provides more accurate information on left ventricular volumes, mass and ejection fraction, which have a better correlation with those measured using CMR [60]. However, three-dimensional echo is still underutilized in AFD, and there are no data about its clinical usefulness to improve the outcome of patients with AFD.

## 5. Cardiac Magnetic Resonance

CMR provides relevant information that contribute to diagnosing, monitoring and treating patients with AFD (Table 3). One of the main advantages of CMR is the possibility to assess both cardiac mass, function and changes in tissue characteristics in a single, non-invasive examination. In comparison with echocardiography, CMR is superior in many aspects. With its high spatial resolution, a significant advantage of CMR in AFD is that it allows for a more precise assessment of LV wall thickness and the extent of hypertrophy. For example, in 48 patients with HCM, CMR was capable of identifying regions of LV hypertrophy not well recognized by echocardiography [62]. CMR was solely responsible for the diagnosis of the HCM phenotype in a significant number of patients with non-diagnostic echocardiography [63]. In 32 AFD patients, Azari et al. showed that echocardiography consistently overestimates LV mass index compared to CMR [64]. Whereas CMR showed an increasing trend over time in LV mass index, cardiac ultrasound failed to identify this trend, which indicates that differences in this parameter may not be detected by echocardiography because of the tendency to always be overestimated and the high variability between measurements [64]. Because patients treated with ERT had a slower LV mass increase than those without therapy [65], CMR could be the preferred technique in AFD, not only for the diagnosis but also for the monitoring of the effectiveness of the treatment. Another interesting aspect is that CMR allows easier measurement of LV papillary muscle mass [38,66]. As mentioned above, the presence of LV papillary muscle hypertrophy is quite common in the advanced stage of the disease, and interestingly, in those cases, the contribution of LV papillary muscle mass to the total LV mass is much higher in AFD than in normal subjects (up to 20% in LVH-positive AFD patients compared to approximately 8% in normal hearts). Thus, the presence of disproportionate papillary muscle hypertrophy could help diagnose ADF in patients without left ventricular hypertrophy [67]. Elevated LV mass, as assessed by CMR, is known to be associated with ventricular arrhythmia, in particular when high trabecular and papillary muscle volume is present [68]. Hence, CMR could be useful to better quantify LV mass as well aspapillary muscle volume for its prognostic value. Deva et al. reported that patients with elevated CMR-indexed left ventricular mass had a greater incidence of ventricular arrhythmia [68]. These data suggest the possibility of a link between high LV mass and ventricular arrhythmia, as also seen in HCM in which significant hypertrophy predicts adverse events [69].

## 6. Tissue Characterization

### 6.1. T1 and T2 Mapping

Tissue characterization is crucial for a better evaluation of AFD. CMR is the primary imaging modality for myocardial tissue characterization. Parametric mapping techniques with CMR allow the quantification of changes in myocardial composition (e.g., glycosphingolipid accumulation in AFD) based on changes in T1, T2 and T2* relaxation times and extracellular volume [70]. Unlike T1-, T2-or T2*-weighted images, mapping permits both visualization and quantification of the disease process, independent of whether myocardial disease is focal or diffuse. Native T1 is low in AFD, unlike what occurs in any other cause of hypertrophy, except for iron overload [69]. T1 shortening is due to sphingolipid accumulation (Figure 5). Thompson et al. have demonstrated that there is a sex difference in T1 values with longer myocardial T1 values in female AFD patients [71]. T1 is low in around half of patients with Anderson-Fabry, even in the absence of hypertrophy. Thus, lower myocardial T1 in AFD patients without evidence of LVH might be useful to detect early cardiac involvement [72]. However, it is important to recognize that segmental pseudo-normalization or elevation of T1 may occur over time in the basal inferolateral wall of the left ventricle, likely reflecting the presence of areas of either mixed storage and fibrosis or inflammation. A significant percentage of patients with AFD have renal dysfunction, the paradoxical increase in T1 values could allow identification of focal fibrosis without contrast injection [73]. The finding of high T2 values observed in the basal inferolateral region, or, in general, in LGE areas, suggests that inflammation may contribute to the pathogenesis of AFD cardiomyopathy. This finding is not observed in HCM or chronic myocardial infarction [74,75]. Moreover, it has been demonstrated that a reduction in T2 relaxation times is correlated with a reduction in LV mass after 45 or 48 months of ERT [76]. Thus, it may be useful to use the time T2 to monitor the progress of the therapy.

### 6.2. Late Gadolinium Enhancement

The presence of fibrosis can be assessed by the amount and distribution of the LGE after gadolinium-based contrast agents administration. Replacement fibrosis is considered a sign of disease progression [77] and may precede the onset of LVH, particularly in females [78,79,80]. Accordingly, by LGE-CMR imaging, fibrosis has been found in both AFD patients with and without LV hypertrophy.

The typical LGE pattern is at the mid-wall in the basal inferolateral area of the left ventricle (Figure 5). Therefore, myocardial late enhancement in infero–postero-lateral region with no affection of the endocardium may be considered a red flag suggestive of diagnosis of AFD (Figure 6) [81]. However, atypical patterns have been found in 1/5 of patients with pathological LGE [68]. In patients with asymmetrical septal hypertrophy and apical hypertrophy, LGE has been described in the basal antero-septum and apical segments, respectively, miming the presence of HCM. In the advanced disease, the LGE can be very extensive and diffuse, with a less specific appearance [68,82].

Hanneman et al. demonstrated that the amount of LGE was strongly correlated with the incidence of cardiovascular events (arrhythmias, severe heart failure, cardiac death), stressing the prognostic value of CMR in risk stratification of AFD patients [83].

### 6.3. Speckle Tracking Analysis in CMR

The use of speckle tracking analysis is less common in CMR than echocardiography. It can be reliably estimated, using image feature tracking methods applied to SSFP CMR images [84].

Few studies have analyzed CMR-derived GLS and GCS in AFD patients. No significant differences in GLS between AFD patients and controls have been found [85]. Regarding GCS, a significant increase in circumferential strain in patients with LVH phenotype has been found, and it may imply the presence of functional cardiovascular impairment [86]. Mathur et al. demonstrated that base-to-apex CS gradient discriminates between AFD patients without hypertrophy or LGE and healthy controls independent of native T1, suggesting that loss of base-to-apex CS gradient may be an early marker of cardiac involvement in AFD [85]. More recently, Augusto et al. described a slightly reduced GLS as a primary cardiac phenomenon, because of the altered myocardial coupling to the systemic vasculature due to systemic endothelial and smooth muscle changes [87]. However, additional studies will be necessary to validate the utility of CMR speckle tracking in AFD patients.

### 6.4. Cardiovascular Magnetic Resonance Perfusion Mapping

As shown by Knott et al. [88], AFD patients have reduced perfusion, particularly in the sub-endocardium with greater reductions in patients with LVH, storage, edema and scar. Perfusion is reduced even without LVH, suggesting it is an early disease marker. The reduction in perfusion implies that AFD patients may have an early microvascular dysfunction thatcould contribute to the progression from storage to fibrosis.

## 7. Nuclear Scintigraphy and Positron Emission Tomography

In the last few years, the emerging role of nuclear imaging techniques in AFD has been studied. Glycosphingolipids’ storage also affects endothelial and smooth muscle cells causing microvascular dysfunction at different levels, including the myocardium. This process results in a global reduction in coronary flow detected by single-photon emission computed tomography (SPECT) imaging with 99mTc sestamibi [89].

Myocardial hypoperfusion can also be estimated using positron emission tomography (PET) imaging. Tomberli et al. demonstrated that a global reduction in coronary flow reserve was an early sign of cardiac involvement, regardless of sex and LVH [90]. ERT therapy seems to not affect coronary microvascular dysfunction [89,91].

Few studies show that PET imaging, and in particular hybrid cardiac PET-MR imaging, is also useful to detect myocardial inflammation, present in an early phase of the disease, and to identify different stages of disease progression. The presence of 18F-FDG uptake correlates with impaired LV longitudinal function at echo [92] and LGE areas [93].

Using hybrid positron emission tomographymagnetic resonance imaging, Imbriaco et al. demonstrated that focal 18F-FDG uptake was associated with a trend towards a pseudo-normalization of abnormal T1 mapping values in female AFD patients that may represent an intermediate stage before the development of myocardial fibrosis, suggesting a potential relationship between progressive myocyte sphingolipid accumulation and inflammation [94]. Hybrid PET-MR imaging studies could play a role in the early detection of cardiac involvement, allowing a very timely and more effective therapeutic approach, well before the development of structural changes and myocardial fibrosis.

Nuclear imaging is useful to study the cardiac effect of autonomic nervous system dysfunction in AFD [95]. The chemical123I-meta-iodobenzylguanidine (MIBG) imaging shows a decreased uptake, indicative of regional myocardial denervation. Decreased uptake correlates with GLS reduction at echo [96] and might precede fibrosis [97,98].

Currently, SPECT and PET imaging are used only for investigation reasons, but they look promising in detecting early cardiac involvement in AFD patients.

## 8. Conclusions

“Fabry disease-often seen, rarely diagnosed” is how Hoffmann and Mayatepek titled their AFD review [99]. Current screening practices likely capture only a small portion of AFD. A greater awareness in the medical community is needed to emphasize that AFD is not a very rare disorder and that it is not uncommon at all in high-risk populations in which screening is usually omitted. Particular attention should be given to patients presenting with kidney damage, cryptogenic stroke, unexplained LVH, gastrointestinal symptoms, hearing impairment, lymphedema, diminished perspirations, acroparesthesias, corneal opacities and angiokeratoma, which are considered clinical markers associated with AFD. AFD should be suspected in patients with a family history or in those who present with the clinical features that suggest the diagnosis. The diagnosis is typically confirmed by enzymatic and/or molecular genetic testing. Regarding the role of cardiac imaging in the management of Fabry patients, it is involved in many aspects: the initial diagnostic suspicion of AFD in case of evidence of unexplained heart damage associated with extracardiac AFD red flags, the differential diagnosis with other cardiomyopathies, the early detection of heart damage in patients with already diagnosed AFD and monitoring its evolution, to allow decisions regarding the initiation of chaperone or enzyme replacement therapy, and to guide its follow-up. The echocardiographic examination is the first-line technique to suspect and manage AFD. However, there are no pathognomonic echocardiographic features of AFD. CMR has emerged as a powerful imaging tool to identify lesions of AFD in patients in whom echocardiography fails to detect relevant LVH or other cardiac damage. Its strength is in characterizing tissue using LGE or T1 and T2 mapping. LGE imaging is the non-invasive gold standard for the evaluation of replacement fibrosis/scarring. The tissue damage highlighted by LGE, initially located in the basal inferolateral wall, has prognostic implications and predicts a lack of response to enzyme replacement therapy. Low native myocardial T1 values could represent a useful, early biomarker of cardiac involvement in AFD, superior to left ventricular hypertrophy and LGE imaging. T2 mapping is sensitive to inflammation. Studies with T2 mapping, supported by histological studies, deny the model of AFD as simple storage cardiomyopathy and have led to the identification of an important role of chronic inflammation in the early progression of the disease. This recognition may have implications on future management strategies including consideration for immunosuppressive therapy in the hope of improving the course of AFD. These observations justify an increasing role of CMR in the routine clinical evaluation of patients with AFD. As with echocardiography, CMR findings in themselves are not diagnostic of AFD and must be considered within the clinical contest of an individual patient and confirmed with enzymatic and genetic analysis.

An integrated multi-modality imaging approach including both echocardiography and CMR might be optimal for the management of AFD patients. In the future, echocardiography, by its large availability and low cost, will remain the initial imaging modality of choice in patients with proven or suspected AFD, but the role of CMR will be likely to increase so much as toalso become an essential diagnostic test in the initial evaluation.

## Figures and Tables

**Figure 1 jcm-10-01994-f001:**
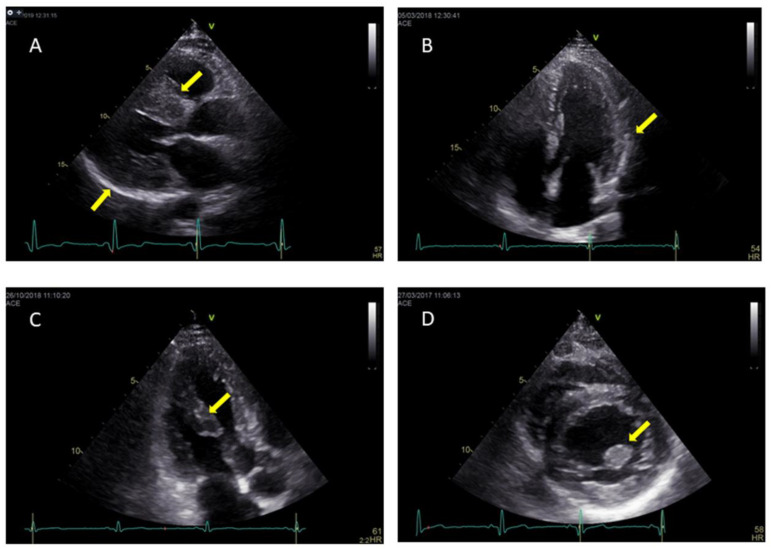
Parasternal long axis shows symmetrical hypertrophy in AFD patient (**A**); echocardiographic binary sign of left ventricular endocardial border in AFD patient (**B**); hypertrophy of papillary muscles in AFD patient: in apical long axis view (**C**) and in parasternal short axis (**D**).

**Figure 2 jcm-10-01994-f002:**
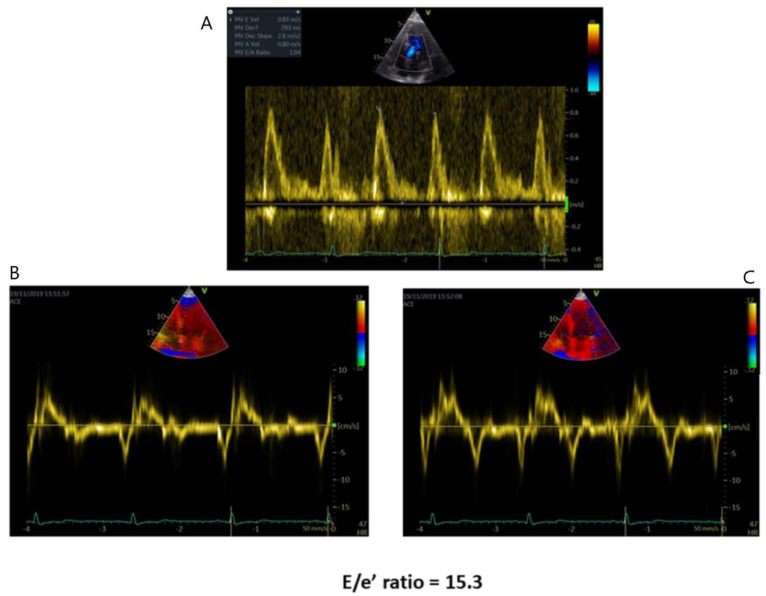
Diastolic dysfunction in AFD patients with preserved LVEF. (**A**) E/A ratio tissue. (**B**) Doppler recordings of septal mitral annular velocities. (**C**) Tissue Doppler recordings of lateral mitral annular velocities.

**Figure 3 jcm-10-01994-f003:**
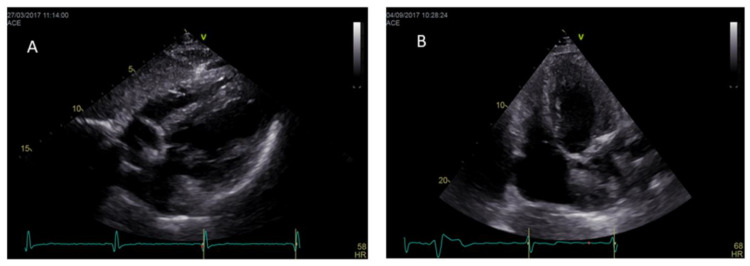
Hypertrophy of both the right and left ventricle in subcostal view (**A**) and in modified apical four chamber view (**B**) in two different AFD patients.

**Figure 4 jcm-10-01994-f004:**
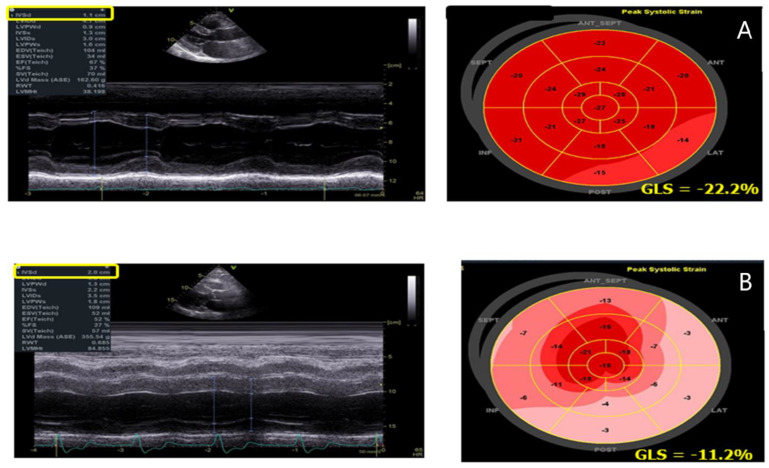
AFD patient with initial septal hypertrophy and reduction in the infero-lateral regions’ strain (**A**); AFD patient with advanced hypertrophy and strain reduction inall basal segments of the left ventricle (**B**).

**Figure 5 jcm-10-01994-f005:**
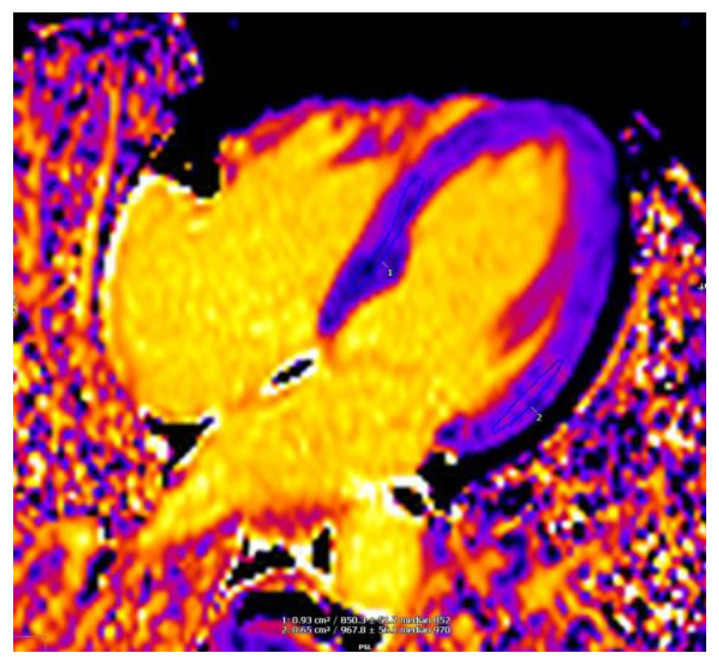
T1 mapping sequence shows a lower T1 time in the posterior interventricular septum in an AFD patient.

**Figure 6 jcm-10-01994-f006:**
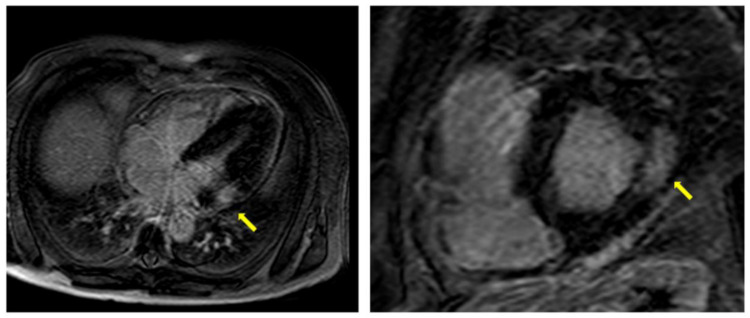
LGE-CMR shows typical LGE pattern at the mid-wall in the basal infero-lateral area of the left ventricle in two different AFD patients (yellow arrow).

**Table 1 jcm-10-01994-t001:** Standard echocardiography in ADF cardiomyopathy.

Standard Echofindings	Description	Features
LV Hypertrophy	-Usuallysymmetric with concentric geometry -Rarely asymmetric septal hypertrophy, apical hypertrophy or eccentric geometry	-Predominant manifestation of AFD cardiomyopathy -Occurs in the 4th decade of life in men, later in women
Binary Sign	-Hyperechogenic endocardial surface adjacent to a hypoechogenic subendocardial layer	-Once considered pathognomonic -Overall low sensitivity and specificity
Prominent Papillary Muscles	-Papillary muscle thickening and hyperechogenicity	-Late sign, not for screening purposes
Preserved LV EF	-LV EF is usually in the normal range	-LV systolic dysfunction is a marker of severe cardiac involvement related to a poor prognosis
Diastolic Dysfunction	-Mitral flow Doppler parameters alteration -E/e’ ratio increase -LA dilation	-In AFD patients with LV hypertrophy, diastolic dysfunction underlies the symptoms of heart failure
Right Ventricle Hypertrophy	-Usually RV systolic function is preserved -Often associated withRV diastolic dysfunction	-Its prevalence varies between studies -No sex differences

LV: left ventricle; AFD: Anderson-Fabry disease; EF: ejection fraction; LA: left atrial; RV: right ventricle.

**Table 2 jcm-10-01994-t002:** Advanced echocardiography in AFD cardiomyopathy.

Advanced Echocardiography	Description	Features
GLS	-Reduction in LV GLS with a prevalent involvement of the infero-lateral wall of the LV	-Correlates with LGE at CMR
GCS	-Reduction in the normal base-to-apex CS gradient	-Differential diagnosis with HCM where GCS increases with a preserved base-to-apex gradient
RVLS	-Reduction in the RV Longitudinal strain	-Early sign of RV dysfunction

GLS: global longitudinal strain; LGE: late gadolinium enhancement; CMR: cardiovascular magnetic resonance; GCS: global circumferential strain; HCM: hypertrophic cardiomyopathy; RVLS: right ventricle longitudinal strain.

**Table 3 jcm-10-01994-t003:** Cardiovascular magnetic resonance in AFD cardiomyopathy.

CMR Sequences	Description	Features
Cine-sequences	-Measurement of LV mass, ventricular volumes, LV and RV EF, wall motion assessment	-Better quantification of LV papillary muscle mass
LGE	-Fibrosis usually localized at mid-wall in the basal infero-lateral area of LV -Very extensive and diffuse in advanced AFD	-Suggestive of AFD when in the typical localization -Additionally present in patients without LVH -Strongly correlated with more CV events
T1 mapping	-Lower native T1 times	-Early sign of cardiac involvement -Pathognomonic of AFD -Pseudo-normalization of T1 times correlates with the presence of LGE
T2 mapping	-Elevation of T2 times in inferolateral wall or LGE areas	-Suggestive of myocardial inflammation -T2 times elevation correlates with troponin elevation -No pathognomonic
ECV	Normal values except in LGE areas	-No pathognomonic -Reflects interstitial fibrosis
Speckle Tracking Analysis	-GLS shows no significant differences -GCS has a significant increase in LVH patients with loss of base-to-apex GCS gradient	-GCS may be an early marker of cardiac involvement -Speckle tracking analysis is not commonly used in CMR, further investigations are required

CMR: cardiovascular magnetic resonance; AFD: Anderson-Fabry disease; LV: left ventricle; RV: right ventricle; EF: ejection fraction; LGE: late gadolinium enhancement; LVH: left ventricle hypertrophy; CV: cardiovascular; ECV: extracellular volume; GLS: global longitudinal strain; GCS: global circumferential strain.

## Data Availability

Data are taken from literature.

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
