# Peer review of "Cardiac Imaging in Anderson-Fabry Disease: Past, Present and Future"

_jcm, 2021, doi:10.3390/jcm10091994_

Round 1

Reviewer 1 Report

Cardiac Imaging in Anderson-Fabry Disease: Past, Present and Future.

Brief summary: This review article aims to give an overview of the use of cardiac imaging in Fabry disease in the past, present and future.

Broad comments:

- The structure of the review article is well organized however the review article focuses too much on echocardiography imaging. Certain aspects of the review do not seem very relevant to the current care of Fabry patients. Instead the author should focus on a few important and clinically useful details and expand them in more detail. For eg:

  1. Line 122 ‘a cut off of 14.8 for the septal E/e’ ratio has shown to be the best predictor of presence of LGE’. Is this really clinically used?

  1. Line 126-128 mentions about atrial damage in Fabry disease and maybe useful in early diagnosis. I’m not sure how relevant or clinically useful this is? Diastolic dysfunction is a characteristic but not specific to Fabry disease.

  1. Line 173-174 What is the significance of the different longitudinal strain reduction patterns seen in Fabry disease?

-The article needs improvement in English language and style of writing. There are also spelling mistakes throughout the article. For eg in Table 2 itself there are spelling mistakes ‘AFD Cardiomyopathy’ ‘differential diagnosis’. Some of the terms used are unusual for example the use of ‘proper medical history collection in line 60, wide court in line 160 and the use of symmetrical hypertrophy rather than concentric hypertrophy’.

-Certain references used are incorrect. Please see details in specific comments.

-More details should be included in other imaging modalities. There have been a lot of new research in CMR for example and clinically has been increasingly used not just in early diagnosis but in other domains of Fabry disease.

-The article is also supposed to be reviewing on the future and novel work as well but it is unclear in the article where the future direction of cardiac imaging is in Fabry disease.

Specific comments:

-line 42 Females can actually get severe symptoms, not just ‘mild to moderate symptoms’ as mentioned by the authors

-line 43-44 ‘Cardiovascular involvement in AFD patients usually occurs in the fourth decade of life.’ This differs in males and females and should be referenced

-line 65-66 This sentence does not make sense ‘In the absence of non-cardiac symptoms and signs, the diagnosis of AFD can be challenging also employing sophisticated imaging techniques.’

-line 69 ‘unexplained myocardial thickening’ is not a feature of AFD

-lines 98-101 should be in figure legend of Figure 1

-lines 104-105. Sudden cardiac death/ventricular arrhythmia and not diastolic dysfunction is the cause of death in Fabry disease.

-Lines 138-139 ‘the prevalence of RVH in 50-70% of all patients with Fabry’ is misleading as it’s proportional to prevalence of LVH. The prevalence of LVH in AFD is less than 50-70%.

-Line 147 Aorta section. Reference 37 is not Barbey et al and AFD has not been included currently in hereditary disease associated with thoracic dilatation/aneurysm

-Line 157 How does mitral valve insufficiency can be induced by papillary muscle prominence?

-Table 1, certain words are merged together and there are spelling mistakes

- ‘3D Echo provides more accurate information on left ventricular volumes, mass, and ejection fraction, which have a better correlation with those measured using CMR (64)’. I don’t think this statement is right and the reference does not mention 3D Echo.

Line 230 ‘in a significant minority of patients with non-diagnostic echocardiography’. I’m not sure what does significant minority mean.

-There should be figures included in other imaging modalities for example low native T1 in a Fabry patient

-Certain sentences need referencing for eg line 82-83 ‘in the advanced phases of the disease, is correlated to a worse prognosis’ and line 236-237 ‘patients treated 236 with ERT had a slower LV mass increase than those without therapy’

-Isn’t elevated T2 in Table 3 pathognomonic if it’s seen in Fabry disease and not in other disease?

Author Response

The structure of the review article is well organized however the review article focuses too much on echocardiography imaging. Certain aspects of the review do not seem very relevant to the current care of Fabry patients. Instead the author should focus on a few important and clinically useful details and expand them in more detail. For eg:

Line 122 ‘a cut off of 14.8 for the septal E/e’ ratio has shown to be the best predictor of presence of LGE’. Is this really clinically used?

Thank you for your valuable comment. The value of septal E/e’ is considered a significant noninvasive marker of increased LV filling pressure. In Fabry patients this sign is an indirect marker of of advanced diastolic dysfunction associated with LGE at CMR (Boyd AC, Lo Q, Devine K, Tchan MC, Sillence DO, Sadick N, Richards DA, Thomas L. Left atrial enlargement and reduced atrial compliance occurs early in Fabry cardiomyopathy. J Am SocEchocardiogr. 2013;26:1415–1423.  doi: 10.1016/j.echo.2013.08.024.). In our experience both measurements are associated with unfavorable prognosis.

Line 126-128 mentions about atrial damage in Fabry disease and maybe useful in early diagnosis. I’m not sure how relevant or clinically useful this is? Diastolic dysfunction is a characteristic but not specific to Fabry disease.

As the reviewer rightly suggests diastolic dysfunction is not specific of Fabry disease, but it does occur in the early stages of the disease often before LVH appears. Since early treatment can improve clinical outcomes it can be important to detect cardiac involvement.

Line 173-174 What is the significance of the different longitudinal strain reduction patterns seen in Fabry disease?

Longitudinal strain correlates with different mechanisms of cardiac damage: glycosphingolipid accumulation, inflammation and immune activation. It seems that the different longitudinal strain reduction patterns correlate with different stages of the disease from the earliest with a normal or near-normal regional LS to the latest with almost global LS reduction

The article needs improvement in English language and style of writing. There are also spelling mistakes throughout the article. For eg in Table 2 itself there are spelling mistakes ‘AFD Cardiomyopathy’ ‘differential diagnosis’. Some of the terms used are unusual for example the use of ‘proper medical history collection in line 60, wide court in line 160 and the use of symmetrical hypertrophy rather than concentric hypertrophy’.

Thank you for the suggestion, we performed an extensive language editing.

Certain references used are incorrect. Please see details in specific comments.

Thank you for the suggestion, we reviewed and corrected the reference list.

More details should be included in other imaging modalities. There have been a lot of new research in CMR for example and clinically has been increasingly used not just in early diagnosis but in other domains of Fabry disease.

Thank you for the suggestion. We have added the latest research especially in terms of CMR application in Fabry disease both in early diagnosis and in follow-up of Fabry disease.  We have reported this in specific sessions (T1 and T2 mapping and Speckle tracking analysis in CMR). Page 9,lines 16-19 and 22-24

The article is also supposed to be reviewing on the future and novel work as well but it is unclear in the article where the future direction of cardiac imaging is in Fabry disease.

We tried to better address the topic in the conclusions. Thanks for the valuable suggestion

Specific comments:

line 42 Females can actually get severe symptoms, not just ‘mild to moderate symptoms’ as mentioned by the authors.

Thank you we totally agree and changed the text.

line 43-44 ‘Cardiovascular involvement in AFD patients usually occurs in the fourth decade of life.’ This differs in males and females and should be referenced.

We deleted lines 43 and 44 and reported in the text (lines 58-59) that in males LVH occurs at younger age than in females.

line 65-66 This sentence does not make sense ‘In the absence of non-cardiac symptoms and signs, the diagnosis of AFD can be challenging also employing sophisticated imaging techniques.’

Thank you for the suggestion we changed the text.

line 69 ‘unexplained myocardial thickening’ is not a feature of AFD

We changed it to “unexplained left ventricular hypertrophy”, thank you.

lines 98-101 should be in figure legend of Figure 1

Thank you we changed it.

lines 104-105. Sudden cardiac death/ventricular arrhythmia and not diastolic dysfunction is the cause of death in Fabry disease.

We agree with your observation. We, therefore, changed the sentence specifying that: “LV diastolic dysfunction occurs early in the AFD disease stages, and represents the substrate for the onset of symptoms and the leading cause of heart failure in these patients” (page 4).

Lines 138-139 ‘the prevalence of RVH in 50-70% of all patients with Fabry’ is misleading as it’s proportional to prevalence of LVH. The prevalence of LVH in AFD is less than 50-70%.

We checked and changed the rates of RVH ranging from 40 % to 71%., as reported in literature (Palecek T, Dostalova G, Kuchynka P, Karetova D, Bultas J, Elleder M, et al. Right ventricular involvement in Fabry disease. J Am Soc Echocardiogr 2008;21:1265-8.;Niemann M, Breunig F, Beer M, Herrmann S, Strotmann J, Hu K, et al. The right ventricle in Fabry disease: natural history and impact of enzyme replacement therapy. Heart 2010;96:1915-9).

Line 147 Aorta section. Reference 37 is not Barbey et al and AFD has not been included currently in hereditary disease associated with thoracic dilatation/aneurysm.

We corrected reference 37. We deleted the sentence: “Consequently, they suggested that AFD should be included in the list of hereditary diseases that are associated with ascending thoracic dilation/aneurysm.”

Line 157 How does mitral valve insufficiency can be induced by papillary muscle prominence? Minor structural abnormalities in mitral valve are frequent. Patients with mitral valve

Thank you for your observation. In patients with asymmetrical septal hypertrophy, mimicking hypertrophic obstructive cardiomyopathy, Linhart et al. noted the typical systolic anterior motion of the anterior mitral leaflet that contributes to mitral valve dysfunction. The mitral valve appears to be affected in relatively young subjects. The valvular structural changes are accompanied by frequent, but mostly non-significant (mild-to-moderate), regurgitation. (Linhart A, Lubanda JC, Palecek T, Bultas J, Karetova D, Ledvinova J, et al. Cardiac manifestations in Fabry disease. J Inherit Metab Dis 2001; 24(suppl 2):75-83.)

Table 1, certain words are merged together and there are spelling mistakes

Thank you we extensively review and edited the manuscript.

‘3D Echo provides more accurate information on left ventricular volumes, mass, and ejection fraction, which have a better correlation with those measured using CMR (64)’. I don’t think this statement is right and the reference does not mention 3D Echo.

Thanks we fixed the error.

Line 230 ‘in a significant minority of patients with non-diagnostic echocardiography’. I’m not sure what does significant minority mean.

Thank you for the suggestion we changed it.

There should be figures included in other imaging modalities for example low native T1 in a Fabry patient

Thank you for the suggestion we added it (Figure 5).

Certain sentences need referencing for eg line 82-83 ‘in the advanced phases of the disease, is correlated to a worse prognosis’ and line 236-237 ‘patients treated 236 with ERT had a slower LV mass increase than those without therapy’

Thank you for the suggestion we added these references. Reference 7 (Patel V, O'Mahony C, Hughes D, Rahman MS, Coats C, Murphy E, Lachmann R, Mehta A, Elliott PM. Clinical and genetic predictors of major cardiac events in patients with Anderson-Fabry disease. Heart 2015;101:961-966.   doi: 10.1136/heartjnl-2014-306782) and 67 (Imbriaco M, Pisani A, Spinelli L, Cuocolo A, Messalli G, Capuano E, Marmo M, Liuzzi R, Visciano B, Cianciaruso B, Salvatore M. Effects of enzyme-replacement therapy in patients with Anderson-Fabry disease: a prospective long-term cardiac magnetic resonance imaging study. Heart. 2009 Jul;95(13):1103-7. doi: 10.1136/hrt.2008.162800.)

Isn’t elevated T2 in Table 3 pathognomonic if it’s seen in Fabry disease and not in other disease?

It is not pathognomonic because indicates only the presence of inflammation that may be present in other inflammatory cardiomyopathies such as myocarditis, sarcoidosis, Chagas disease, etc. (Messroghli, Daniel R et al. “Clinical recommendations for cardiovascular magnetic resonance mapping of T1, T2, T2* and extracellular volume: A consensus statement by the Society for Cardiovascular Magnetic Resonance (SCMR) endorsed by the European Association for Cardiovascular Imaging (EACVI).” Journal of cardiovascular magnetic resonance : official journal of the Society for Cardiovascular Magnetic Resonance vol. 19,1 75. 9 Oct. 2017, doi:10.1186/s12968-017-0389-8)

Reviewer 2 Report

  1. The incidence of Fabry disease should be corrected. Recent papers, especially those based on newborn screening show that the incidence is about 1:10,000 live births.
  2. LVEF is often normal but on occasion it is above normal. The heart is often described as hyperdynamic.
  3. Change gender to sex.
  4. Figure legends should be improved so people who are not cardiologists can understand. For example Figure 2, it not clear what is shown there.
  5. There are number of typographical errors that need corrected.

Author Response

The incidence of Fabry disease should be corrected. Recent papers, especially those based on newborn screening show that the incidence is about 1:10,000 live births.

Thank you for the observation we changed the text

LVEF is often normal but on occasion it is above normal. The heart is often described as hyperdynamic.

Thank you, we changed the text accordingly.

Change gender to sex.

We modified it. Thank you

Figure legends should be improved so people who are not cardiologists can understand. For example Figure 2, it not clear what is shown there.

Thank you, we improved the quality of the figure description as you suggested.

There are number of typographical errors that need corrected.

Thank you we extensively review and edited the manuscript.

Round 2

Reviewer 1 Report

Thank you for the reply. The review article is now improved. There are still some minor spelling mistakes otherwise no further comments.